# The Black Hole: CAR T Cell Therapy in AML

**DOI:** 10.3390/cancers15102713

**Published:** 2023-05-11

**Authors:** Erden Atilla, Karim Benabdellah

**Affiliations:** 1Fred Hutchinson Cancer Research Center, Clinical Research Division, 1100 Fairview Ave N, Seattle, WA 98109, USA; 2GENYO Centre for Genomics and Oncological Research, Genomic Medicine Department, Pfizer/University of Granada/Andalusian Regional Government, Health Sciences Technology Park, 18016 Granada, Spain; karim.benabdel@genyo.es

**Keywords:** acute myeloid leukemia, cellular therapies, chimeric antigen T cells, T cell receptor T cells, CAR NK cells, RNAseq

## Abstract

**Simple Summary:**

Unlike B cell malignancies, the progress on generating an adoptive T cell therapy for relapsed/refractory acute myeloid leukemia (AML) remains insufficient. This review focuses on the main challenges in the field and novel strategies to overcome them.

**Abstract:**

Despite exhaustive studies, researchers have made little progress in the field of adoptive cellular therapies for relapsed/refractory acute myeloid leukemia (AML), unlike the notable uptake for B cell malignancies. Various single antigen-targeting chimeric antigen receptor (CAR) T cell Phase I trials have been established worldwide and have recruited approximately 100 patients. The high heterogeneity at the genetic and molecular levels within and between AML patients resembles a black hole: a great gravitational field that sucks in everything. One must consider the fact that only around 30% of patients show a response; there are, however, consequential off-tumor effects. It is obvious that a new point of view is needed to achieve more promising results. This review first introduces the unique therapeutic challenges of not only CAR T cells but also other adoptive cellular therapies in AML. Next, recent single-cell sequencing data for AML to assess somatically acquired alterations at the DNA, epigenetic, RNA, and protein levels are discussed to give a perspective on cellular heterogeneity, intercellular hierarchies, and the cellular ecosystem. Finally, promising novel strategies are summarized, including more sophisticated next-generation CAR T, TCR-T, and CAR NK therapies; the approaches with which to tailor the microenvironment and target neoantigens; and allogeneic approaches.

## 1. Introduction

Since the early studies in the eighteenth century on the development of the first vaccines, researchers have attempted to eliminate tumors by harnessing the immune system. One strategy, adoptive cell therapy, uses T cells that can recognize tumor antigens through tumor-specific receptors. Since the 1980s, chimeric antigen receptor T cells (CAR T cells) have revolutionized the treatment algorithms of patients with lymphoma, acute lymphoblastic leukemia, and multiple myeloma [1]. They have shown dramatic success in the clinic, improving survival and quality of life for patients that would otherwise reach the end of care with conventional therapies. Currently, six CAR T cell products are approved by the United States Food and Drug Administration (FDA), and approvals are expanding to Europe and many other countries around the world.

There are distinct considerations in acute myeloid leukemia (AML), the most common acute leukemia in adults. AML is an aggressive blood cancer characterized by a collection of immature cells of myeloid lineage that exhibit partial or complete arrest of maturation [2]. The heterogeneity and intrinsic variability of the tumor make patient responses hard to predict, and around 75% of patients ultimately relapse. Treatment resistance (10–40%) and relapse remain the major consequences during disease follow-up, highlighting the urgent need for novel therapeutic approaches [3]. Allogeneic hematopoietic stem cell transplantation (allo-HSCT) is the only curative option, but many patients are not suitable candidates [4].

Many in vitro and in vivo studies have shown that CAR T cells against surface proteins, such as CD33, CD123, CLL-1, CD13, CD7, NKG2D ligand, CD38, CD70, and TIM3, effectively eradicate AML cells [5]. However, the clinical trials are limited, with not very promising response rates accompanied by high ‘on-target off-tumor’ toxicity due to the frequent expression on healthy hematopoietic stem cells or progenitors, as well as other tissues. The major clinical trials and case reports are shown in Table 1; ongoing trials were summarized in a review by Marofi et al. [6]. The data in the CAR T field for AML so far resemble a black hole with a bulky mass which is cumulatively increasing and condensing day by day. This mass exerts an immense gravitational pull even light cannot escape; this is similar to the situation with CAR T cells for AML, in that none of the strategies have the power to cure. This review focuses on the main questions regarding the challenges for adoptive immune therapies in the setting of relapsed/refractory AML and the novel approaches to overcome them.

## 2. Challenges in Adoptive T Cell Therapy for Acute Myeloid Leukemia

### 2.1. AML Is Highly Heterogenous

Normal hematopoietic stem cells give rise to mature cells of the myeloid, lymphoid, and erythroid/megakaryocyte lineages. Single-cell RNA sequencing (scRNA-seq) analyses have shown that normal hematopoietic stem cell (HSC) commitment proceeds through a series of increasingly lineage-committed progenitor states [13]. AML consists of leukemia stem cells (LSCs) and differentiated cells. LSCs sustain the disease and display self-renewal, quiescence, and therapy resistance. Differentiated AML cells that lack stem cell characteristics affect tumor biology through pathologic effects on the tumor microenvironment [14].

AML is a highly heterogeneous disease between patients due to the presence of specific chromosomal abnormalities, gene mutations, or gene fusions. Seventeen genetic subtypes have been discovered so far in ELN risk stratification, but the number of molecular entities may increase over time [15]. Not all gene expression subtypes correlate well with the underlying disease-driven gene fusions or mutations, in contrast to acute lymphoblastic leukemia (ALL), which is characterized by distinct gene expression subtypes [16]. The recurrent somatic mutations are categorized by their biological roles, such as signaling and kinase pathway genes. The mutation in FLT3 (which encodes a receptor tyrosine kinase) and KRAS/NRAS (which encode a small GTPase) lead to uncontrolled cell growth and proliferation, whereas the mutation in JAK2 (which encodes tyrosine kinase) promotes leukemogenesis. There are epigenetic modifiers that encode a DNA methyltransferase enzyme (DNMT3A), isocitrate dehydrogenase enzymes (IDH1/2), and a DNA demethylase (TET2). The mutation in ASXL1 (a polycomb repressive complex of proteins implicated in chromatin) can lead to the alteration of DNA methylation patterns, dysregulation of gene expression, and altered hematopoietic differentiation. The dysregulation of transcription factors (e.g., CEBPA, RUNX1, MLL, EVI1, etc.); RNA splicing factors (e.g., SRSF2, which encodes a serine and arginine-rich splicing factor, U2AF1, SF3B1, ZRSR2, etc.); tumor suppressors (TP53); nucleophosmin (NPM1); and cohesin complex genes (e.g., RAD21, STAG2, SMC1A, SM3, etc.) can lead to impaired differentiation and uncontrolled cell growth [17]. The initiating leukemogenic NPM1, TET2, and SMC1A mutations emerge in self-renewing cells that Jan et al. identified as highly purified HSCs, namely ‘pre-leukemic HSCs’ [18]. The same mutations may lead to a highly variable preleukemic burden in different patients [19]. In a study conducted by Stengel and colleagues, 37% of 572 AML cases had fusion events, and 41% of these fusions were detected in cases with TP53 alterations or complex karyotypes [20]. A subset of germline mutations has also been shown in myeloid neoplasms: mutations in GATA 1 gene for Down syndrome patients; DNA damage repair for Fanconi anemia; telomere maintenance genes (DKC1, TERC, TIF2) for dyskeratosis congenita; RUNX1, ANKFRD26, and ETV6 for platelet disorders; ANKRD26 for abnormal thrombopoiesis; and DDX41 for familial myelodysplastic syndrome and AML [21,22]. Epigenetic modifications such as histone modifications, DNA methylation, and post-transcriptional regulation of mRNAs by noncoding RNAs have additional roles in the pathogenic heterogeneity [22].

Furthermore, a crucial problem in patients with relapse or treatment resistance is intratumoral heterogeneity, formed with different subclones of leukemia cells, with distinct genetic and epigenetic features coexisting within a single patient [23,24]. Highly heterogeneous LSCs have variable drug sensitivity. Some LSCs acquire quiescence, which plays a major role in drug resistance profiles [25,26]. Recent advances in genomic, transcriptomic, epigenomic, and proteomic data have helped to elucidate the biological differences between pre-treatment AML cells and their equivalents in relapse. During disease progression in t(8;21) positive AML patients, heterogenous cell populations show their own clonal characteristics and obtain cloned components [27]. In a single-cell analysis, Li et al. found that proliferating stem/progenitor-like cells were reprogrammed to a quiescent stem-like expression pattern in primary refractory AML by upregulation of CD52 and LGALS1 expression [28]. Stratmann and colleagues used mass-spectrometry-based in-depth proteomics to show that at relapse, energy metabolism is reprogrammed by the enrichment of mitochondrial ribosomal proteins and subunits of respiratory chain complexes as well as higher levels of granzymes and lower levels of the anti-inflammatory protein CR1/CD35 [29].

### 2.2. There Is No Ideal Surface Antigen to Target

CAR T cells can bind to cell surface molecules without requiring any antigen processing or HLA expression [30]. The choice of which surface antigen to target is the critical step in manufacturing. The most important feature of an ideal target is the unique and high expression profile on tumor cells, above the detection and activation threshold for CAR T cells, as well as tolerable or no expression on healthy tissues to prevent toxicity. CD19 is now a widely accepted B lineage target of lymphoma and leukemia that is expressed in all tumor cells and absent in normal HSCs as well as all normal tissues [31]. Despite many discoveries related to the immunopathology of AML, a single AML-specific target has still remained elusive. Various surface proteins have been reported as potential targets, such as CD123, CLL-1, CD33, CD44, CD96, CD47, CD23, TIM3, CD7, and FLT3 [32,33,34,35]. However, the considerable risk of on-target/off-tumor activity needs to be addressed. Most of the surface antigens on AML blasts are co-expressed by other healthy tissues, mature myeloid cells, and HSCs, raising concerns about prolonged myelosuppression (Table 2). In some Phase I trials, substantial toxicities and deaths have already been reported [10,36].

CD33 is a common transmembrane protein of the sialic acid-binding immunoglobulin-like lectin (SIGLEC) family found on normal progenitor cells, myeloid cells, monocytes, tissue-resident macrophages and more than 90% of leukemic blasts [37]. Severe pancytopenia and cytokine release syndrome (CRS) were reported to be the main adverse effects after administration in a Phase 1 trial [10]. CD44V6 CAR T has shown promising preclinical results; however, monocytopenia was observed due to the shared expression of CD44v6 in circulating monocytes [38]. CD123, which is the IL-3 receptor subunit, is overexpressed on LSCs, AML blasts, and early hematopoietic cells such as the hematopoietic stem/progenitor cells that induce myeloablation [39]. A recent established target, folate receptor 1 (FOLR1), overexpressed in AML, was shown to have no impact on normal HSPCs in C/G positive pediatric acute megakaryoblastic leukemia, but it is expressed in various healthy tissues (kidney, intestine, lung, retina, placenta, and choroid plexus) and severe lung toxicity has already been demonstrated in studies with a T cell bispecific antibody against FOLR1 in nonhuman primates [40].

### 2.3. Interactions in the Tumor Microenvironment

LSCs reside in a specialized niche that promotes their survival and chemoresistance, through which they can alter their microenvironment [41]. LSCs secrete pro-angiogenic VEGF and interleukins to stimulate angiogenesis to provide additional nutrients, oxygen, and growth factors and to promote proliferation [42,43]. AML blasts support a low-arginine microenvironment [44]. AML LSCs induce the expression of a growth-arresting protein, GAS6, in BM stromal cells [45]. AML cells promote the expression of immunomodulatory factors that impair cytotoxic T lymphocyte (CTL) activation in the tumor microenvironment [46]; these factors include programmed death receptor (PD-1), transforming growth factor β (TGF β), arginase II, prostaglandin E2 (PGE2), cytotoxic T lymphocyte-associated protein 4 (CTLA-4), lymphocyte activation gene 3 (LAG3), and T cell immunoglobulin and mucin-containing-3 (TIM3) on T cells [47]. Furthermore, leukemia cells modulate the NK cell receptor repertoire that inhibits NK cell activity [48].

The interactions are reciprocal: the niche cells also foster LSC growth. In vitro 3D bone marrow microenvironment models have elucidated these interactions [49]. The AML microenvironment contains various cell types, including myeloid-derived suppressor cells (MDSCs), regulatory T cells (Tregs), macrophages, and dendritic cells, that suppress T cell activity. The high expression of Indoleamine 2,3-dioxygenase (IDO) has been reported to promote Treg conversion and enhance the immunosuppressive ability [50]. Treg cells were shown to express abnormally high levels of CD39, and the increase in CD73 has been associated with poor prognosis (E1). The decrease in CXCL12 expression in BM stromal cells triggers the proliferation of AML cells [51]. Osteoblasts in the BM produce WNT ligands to promote leukemia cell survival [52]. AML blasts induce monocytes to secrete pro-inflammatory cytokines, including tumor necrosis factor-α (TNF-α), IL1β, and IL6, and the anti-inflammatory cytokine IL10 [53].

Assessment by scRNA-seq, the cellular indexing of transcriptomes and epitopes by sequencing (CITE-seq), or single-cell assay for transposase-accessible chromatin (ATAC) sequencing has paved the way for the study of cellular heterogeneity and intercellular hierarchies and for the obtaining of insights into the cellular ecosystem of malignant and normal cells. Galen et al. showed that primitive AML cells had dysregulated transcriptional programs with the co-expression of stemness and myeloid priming genes and that differentiated monocyte-like AML cells suppressed T cell activity in vitro by immunomodulatory genes [54]. Furthermore, recent advances in syngeneic models with a fully functional immune system provided an opportunity to evaluate the important aspects of cancer disease progression and the interactions between tumor microenvironments [55,56].

## 3. Promising Strategies to Overcome Challenges

### 3.1. Safer Targets with Less ‘On-Target Off-Tumor’ Effect

Some potential targets appear more reliable in terms of off-tumor toxicity, especially on normal hematopoietic cells, but none of them has proven to be an ideal target in AML. C-type lectin-like molecule-1 (CLL-1) is a type II transmembrane glycoprotein overexpressed in over 90% of AML patients on AML blasts, LSCs, and differentiated myeloid cells that are absent in normal CD34+CD38- hematopoietic stem cells. Tashiro and colleagues reported that CLL-1 CAR T cells can eliminate mature normal myeloid cells but spare healthy HSCs in vitro [57]. The CAR T cells targeting CLL-1 were optimized and demonstrated efficient cytotoxicity in vitro and in vivo [58]. One step ahead of this, CLL-1 CAR T cells were transduced with a second vector encoding soluble IL 15, a cytokine that promotes the survival and proliferation of memory T cells. The expansion and the maintenance of a less differentiated phenotype was detected. In xenograft models of AML, the addition of IL15 induced the production of tumor necrosis factor-alpha (TNFα), likely through the activation of the JAK-STAT pathway. The production of TNF-a contributed to the development of cytokine release syndrome (CRS), and this was controlled with the related antibody blockage [58]. In fact, the TNFα pathway resembles a double-edged sword in immune regulation. The suitable modifications in TNFα signaling may enable the enhancement of CAR activity [59].

CD70, a ligand for CD27 identified as a type II transmembrane glycoprotein, was reported to be expressed on AML bulk cells and leukemic stem cells but not on normal hematopoietic stem cells. It showed a promising anti-tumor effect without toxicity on healthy HSCs [60]. Later, Leick and colleagues designed a panel of CD8 hinge and transmembrane-modified CD70 CAR T cells that were less prone to cleavage and had enhanced binding avidity, leading to more potent activity and expansion [61]. CD7 is a transmembrane glycoprotein that plays a co-stimulatory role in B and T cell lymphogenesis expressed by T cells, NK cells, myeloid progenitors and leukemic cells but not by healthy myeloid cells [62]. The obstacle of shared expression on T cells, which causes T cell fratricide, can be overcome by removing the CD7 gene by CRISPR/Cas9, as shown by our group [63]. Interleukin receptor accessory protein (IL-1RAP) is another promising target on the surface of the LSCs of AML, myelodysplastic syndrome, and chronic myeloid leukemia (CML) but not on healthy HSCs, and it was shown to be effective in vitro and in vivo [64]. PRAME is an intracellular cancer and testis antigen highly expressed in acute myeloid leukemia blasts and normal reproductive tissues. A T cell receptor (TCR) mimic antibody, Pr20, that recognizes the peptide–HLA complex can target intracellular PRAME. Kirkey and colleagues generated PRAME^mTCR^ CAR T cells that were cytotoxic to HLA-A2 restricted AML cells in vitro and in vivo without an impact on normal hematopoiesis [65]. Jetani et al. suggested that Siglec-6 could be a convenient target, sparing HSC or hematopoietic progenitor cells [66].

The efforts to discover efficient targets for immunotherapeutic strategies accelerated following technical progress in proteomic and transcriptomic assays. These assays helped in the understanding of cellular behavior on the protein level instead of immunophenotyping malignant cells. Hoffman et al. described the mass-spectrometry-based phenotyping of HL60 and NB4 cell lines [67]. Perna et al. performed surface-specific proteomic and transcriptomic studies in AML patients and normal tissues to indicate a potential therapeutic target. However, none of the surface proteins showed a similar expression profile to that of CD19. These studies suggested a combinatorial targeting strategy, which was discussed further in other sections [35]. Kohnke and colleagues aimed to discover de novo targets using cell-surface capture technology to detect the surfaceome, a set of proteins expressed on the surface of primary AML patient samples, including surface receptor, transporters, and adhesion molecules, among others. They identified three promising targets: CD148, ITGA4, and Integrin beta-7. Among these, Integrin beta-7 was the most favorable due to low or absent expression in healthy hematopoietic tissues [68]. In a recent scRNA-sec approach, two antigen targets—CSF1R and CD86—revealed potential targets for CAR T cell therapy with broad expression on AML blasts, accompanied by minimal toxicities toward relevant healthy cells and tissues [69].

### 3.2. Limiting the ‘On Target-Off Tumor’ Effect

Unacceptably severe or prolonged toxicities, especially cytopenias and infections, can occur more frequently than in clinical trials with CD19 CAR T cells in lymphomas due to older and therapy-resistant patient populations. It is crucial to maintain anti-cancer immune surveillance and clinical efficacy while avoiding toxicity. Safety or suicide genes are widely used to alleviate the toxicity from CAR T cells [58,70,71]. Alternatively, the administration of cross-reactive CAR T cells can be a bridge to allogeneic stem cell transplants. Tasian et al. showed three different approaches (anti-CD123 messenger RNA electroporated CAR T cells, administration of alemtuzumab, and administration of rituximab to CD20-coexpressing CART123) to eliminate CD123 CAR T cells without effecting the anti-tumor activity in murine models [72].

Many cell engineering approaches have attempted to improve safety when designing CARs, such as with the logic gating of T cell recognition and SnyNotch receptors [73]. Mutation in the anti-CD123 CAR antigen binding domain reduced the antigen binding affinity, as reported by Archangeli et al. [74]. Mild adverse events were demonstrated in the interim analysis of a phase 1 trial of rapidly switchable universal CAR T (UniCAR) targeting CD123 [75,76]. Benmebarek and colleagues generated controllable CAR platform-synthetic agonist receptor (SAR) T cells that were only activated in the presence of their CD33 or CD123 scFv construct in vitro and in AML xenograft models [77]. Dimerizing agent–regulated immunoreceptor complex (DARIC) is a split receptor design that modulates CAR T cell activation by rapamycin to dimerize units. This approach aims to aid hematopoietic recovery and mitigate toxicity. Cooper and colleagues developed a lentiviral DARIC construct that targets a C2 splice isoform with the membrane proximal domain of CD33, and a Phase I study using this strategy is now open for enrollment [78]. We have recently demonstrated the potential benefits of DOX-inducible CAR T therapy, allowing the control of CAR T using an external trigger. It is an effective and sensitive way to turn CAR T activity ON or OFF in order to prevent unwanted side effects and reduce prolonged toxicities [79]. Potentially, the most feasible approach is to knock out the targeted antigen in normal marrow cells. CD33 deletion in primary HSPCs maintained their full function in terms of engraftment and differentiation, and it efficiently reduced off-tumor targeting while preserving on-tumor efficacy [80,81].

### 3.3. Combinatorial Antigen Targeting for Heterogeneity

Combinations of CARs against different AML targets might be a promising solution due to the lack of a leukemia-specific target antigen [73]. Previously, promising results were reported with dual or tri-specific CAR T cells against B cell malignancies and solid tumor models to overcome the heterogeneity and antigen escape [82,83]. Indeed, the phenomenon of ‘antigen escape’ as a reason for the failure of CAR T cells for AML has not been clearly demonstrated in the previous pre-clinical reports [84,85]. Nonetheless, regardless of antigen expression levels, dual-targeting CAR T cells were associated with increased T cell activation and proliferation. This effect might be due to increased interaction with the target cell, favoring immune synapse formation and subsequent T cell recruitment [10,86]. How to combine suitable pairs of antigens to enhance therapeutic efficacy without increasing off-tumor toxicity is a piece of the puzzle that has yet to be solved. Perna et al. suggested four possible combinatorial pairings (CD33+ADGRE, CLEC12A+CCR1, CD33+CD70, and LILRB+CLEC12A) to target AML with an algorithm integrating proteomics and transcriptomics [35].

The combinatorial antigen-targeting strategy has already been applied in AML. Bicistronic CD123 and CD33 CAR T cells showed significant anti-tumor activity in artificially created cell lines (CD33+CD123-, CD33-CD123+) and in vivo [87]. In a phase I trial, CLL-1 and CD33 bicistronic CAR T cells reported remarkable results [88]. Similarly, 10 of 11 pediatric R/R AML patients infused with CLL-1 or CLL-1-CD33 dual CAR T cells had a response (5 reached CR/MRD-) without dose-limiting toxicities [89]. Atilla et al. reported that dual targeting with either a CD33 CAR or a CD123 CAR and a CLL-1 CAR increased anti-tumor activity most profoundly when the target antigen expression on the tumor cells was low. Since the expression of each target antigen is highly variable, we chose to modify T cells with two separate vectors targeting CLL-1 and CD33/CD123 to obtain a mixed product rather than a bicistronic or tandem CAR design in which the molar ratio of each target is fixed [84]. A universal CAR T cell platform (on/off switching mechanism) successfully targeted CD33 and CD123-positive AML blasts in vitro and in vivo [90]. Haubner et al. presented a novel combinatorial ADGRE-2-targeting CAR and a CLEC12A-targeting chimeric costimulatory receptor (CCR) (IF-BETTER gate) that triggered high anti-leukemic activity in vitro and in vivo while sparing vital normal hematopoietic cells [91].

### 3.4. Neoantigens

Neoantigens are limited to malignant clones that arise from somatic mutation [92,93]. Because recurrent gene alterations can be shared by AML patients, neoantigens trigger potent anti-leukemic responses [94]. Distinct from other cancers, AML presents with low mutational burden; so, recognizing the neoantigens arising from mutations is rare [95,96]. Neoantigens from driver gene mutations appear to be ideal targets for immunotherapy since immune evasion is unlikely [93].

The major difficulty in identifying neoantigens was resolved following MS HLA-ligand profiling with whole-exon sequencing and RNA-sequencing techniques [97]. An adequate number of T cells may not be present in AML patients following chemotherapy, or the response may be scarce due to illness-related immunodeficiency [98]. Therefore, the preferred source of neoantigen-specific T cells is usually healthy donors; cells can be isolated using MHC multimers or tetramers carrying the same neoepitopes [99]. Several studies tested the efficacy of neoantigens: CD 8 T cell clones induced by a nonameric neo-peptide (REEMEVHEL) derived from CBFβ-SMMHC fusion protein and in an HLA-A 40:1 restricted manner [98] showed cytotoxicity in in vitro and in vivo models. A neoepitope from NPM1c (CLAVEEVSL) was identified from HLA-A02:01^+^ AML patients, and the healthy CD8 T cells showed lysis when transduced with the same TCR [100]. CD 8+ T cells from an FLT-ITD-positive patient (HLA-A 01:01-restricted) showed an anti-tumor response to a neoepitope (YVDFREYEYY) encoded by the ITD protein region [101].

### 3.5. T Cell Receptor (TCR) T Cells for Treatment of AML

T cell receptor (TCR) engineered T cells act through their modified TCRs and the tumor-associated antigens (TAAs) presented by human leukocyte antigen (HLA) molecules on the surfaces of leukemic cells. The target protein can be expressed intracellularly or on the cell surface. TCR-T cells have less stringent antigen requirements for T cell activation than CAR T cells [102]. TCR-T cell immunotherapy in AML still has barriers that need to be addressed. The major drawbacks are that TAAs might be expressed by non-malignant cells causing on-target, off-tumor toxicities, dose-related toxicity, limited persistence, and the chance of immune escape [103,104]. The dose optimization of TCR-T cells, combining the treatment with exogenous cytokines (e.g., IL-21, IL-7 and IL-15), or adding genetically engineered signaling during cell expansion and demethylating agents such as decitabine might overcome the disadvantages of TCR-T cell application [105]. One other limitation of TCR transfer is the mispairing of endogenous and exogenous TCR components that impair the function; this limitation might be prevented by swapping the constant regions of mouse and human TCRs or codon-optimized cysteine-modified TCRs in which TCR-α and -β are linked by a T2A sequence [106,107,108]. Another approach uses TCR-like CAR T cells that contain scFv and CAR signaling mechanisms that recognize peptides in the context of MHC class I molecules [109].

TCR-T cells against WT1, PRAME, and HA-1 demonstrated anti-leukemic effects in vitro and in a clinical setting in AML [110,111,112,113]. The responses from the clinical trials are variable due to different patient populations, administration of doses, and targets. In the first-in-human trial of TCR-T reacting with WT1 in the context of HLA-A*24:02, only two patients out of eight showed transient decreases in blast counts in bone marrow. The patients had minimal toxicities, including fever, edema, arthritis, and skin reactions [111]. Preferable adverse events were observed in ten patients treated with autologous WT-1-specific TCR-T cells that persisted through 12 months in another trial [114]. More clinical trials on TCR-T cells for AML are detailed in Table 3.

In order to promote graft-versus-leukemia (GVL) reactivity after HLA-matched allogeneic stem cell transplantation, Chapuis et al. isolated WT-1 TCR (TCR_C4_) from HLA-A2^+^ normal donor repertoires (WT-1 TCR CAR T cells) following allogeneic hematopoietic stem cell transplantation (allo-HSCT), inserted TCR_C4_ into Epstein–Barr virus-specific donor CD8^+^ T cells to minimize graft-versus-host disease and infused them prophylactically post-allo-HSCT. This strategy achieved 100% relapse-free survival at a median of 44 months [110]. In a similar approach, donor-derived EBV and/or CMV-specific T cells were redirected by HA-1H TCR to treat HA-1H-positive HLA-A* 02:01-positive patients with high-risk leukemia after allo-HSCT. However, the overall feasibility and efficacy was too low to warrant further clinical development [115].

### 3.6. CAR NK Cells

NK cells are lymphoid cells involved in the innate immune response; they are programmed to kill virus-infected and malignant cells without causing significant graft-versus-host disease, CRS, or neurotoxicity [116]. AML has been an attractive target for NK cell therapy as an allogeneic product [117]. Despite several manipulations for the longer persistence of NK cells, the response to NK cell infusions varies without long-term remissions [117]. NK cells differentiated into cytokine-induced memory-like NK cells following stimulation with IL-12, IL-15, and IL-18 and showed a distinct transcriptional and surface proteomic profile as well as enhanced functionality [118]. Cytokine-induced memory-like (CIML) NK cells were generated by in vitro pre-activation with IL-12, IL-15, and IL-18 and showed promising responses in a Phase I trial in relapsed/refractory AML [119]. Dong et al. reported potent antileukemic activity in vitro and in vivo with peripheral blood-derived CIML NK cells with TCR-like CAR specifically for NPM1c^+^ HLA-A*0201^+^ AML [120]. Genome editing of NK cells to upregulate the cytotoxicity by knocking out the suppression-associated markers ADAM17 (involved in cleavage of CD19) and PD-1 is the next step [121].

The first CAR NK cell therapy was administered for B cell relapsed/refractory lymphoma and CLL. Umbilical-derived, HLA-mismatched, anti-CAR 19 CAR NK cell therapy has been studied in relapsed/refractory lymphoma and CLL and has shown promising results, with 73% of the patients in complete remission [122]. CAR NK cell therapy is favorable in terms of the minimal risk of toxicity and the potential ‘off-the-shelf’ application. The successful application of CD33-targeted CAR-modified NK cells by transduction of blood-derived primary NK cells showed promising cytotoxicity with unimpeded proliferation in vitro and in vivo without observable side effects [123]. The transgenic expression of secretory IL-15 promoted anti-AML activity and enhanced the persistence of CAR NK cells in vitro, but it was associated with systemic toxicities in vivo with anti-CD123 CAR NK [124]. Off-the-shelf cord-derived FLT3 CAR NK cells expressing soluble IL-15 enhanced cytotoxicity and IFN-γ secretion in vitro and improved survival in vivo without HSC toxicity [125]. In a first-in-human Phase I trial, 10 relapsed/refractory AML patients received anti-CD33 CAR NK cells; six of them achieved minimal residual disease-negative CR at day 28 without major toxicities [126]. Other phase I trials on CAR NK cells targeting relapsed/refractory AML are still ongoing (NCT05092451, NCT02892695, NCT02944162).

### 3.7. Manipulations in Manufacturing

AML is a highly aggressive disease affecting older populations. In a relapsed/refractory setting, patients receive many lines of immunosuppressive therapies prior to apheresis, which affects T cell function, and the timeline of manufacturing raises serious concerns in practice. Optimal CAR construct design will preserve the naïve and central memory phenotype as well as the persistence of T cells. It has been shown that naïve and early memory T cells have been enriched by decitabine administered with CD123 CAR T cells [127].

The administration of off-the-shelf ready-to-use products (allogeneic CAR T cells) generated from healthy donors will provide a valuable solution since the CAR T cell products are pre-manufactured without the need for customized manufacturing for a specific patient. Two major issues of allogeneic production—graft-versus-host disease and alloreactivity—can be overcome by various strategies [128]. While the AML off-the-shelf CAR T cell therapies are being developed, these approaches are still in the early stages of developments, in comparison with B cell malignancies. Two patients from a CD38-targeted CAR T cell trial received a donor-derived CAR construct [11]. TCRαβ negative T cells manufactured from healthy donors by TALEN gene editing targeting CD123 (allogeneic, UCART123) eliminated AML in vitro and in vivo with modest toxicity to normal hematopoietic stem/progenitor cells [129]. A phase I clinical trial was halted following the death of the first patient because of severe CRS and capillary leak syndrome with unrelated donor-derived allogeneic anti-CD123 CAR Ts. This trial resumed following the revision of the eligibility criteria and dose modifications [130].

DNA transposon systems are sophisticated systems for stable genetic modification that can deliver large genetic cargos and can be used to reduce cost [131]. Clinical-grade CAR T cell products using Sleeping Beauty and piggyBac for multiple myeloma and leukemia are under investigation [132,133,134]. Gurney and colleagues applied a non-viral approach to primary CAR NK cell production combining the TcBuster DNA transposon system targeting a C-type lectin-like molecule-1 (CLL-1/C-Type Lectin Domain Family 12 Member A, CLEC12) with a GMP-grade Epstein–Barr virus-transformed lymphoblastoid feeder cell (EBC-LCL) for expansion. This approach knocked out a negative regulator of NK cell stimulation, cytokine-inducible SH-2-containing protein (CISH), using CRISPR/Cas9 to enhance the functionality of CLL-1 CAR NK cells without requiring IL-15 stimulation [135].

### 3.8. Strategies to Overcome the Negative Effects of Microenvironment

There are several approaches described to modulate an immunosuppressive microenvironment. Immune evasion such as that with upregulating immune checkpoint proteins has proven to be a way which can dampen the anti-tumor response and limit the efficacy of CAR T cell therapy. Although immune checkpoint blockade in AML has not proven beneficial, [136,137,138] there may still be additional effects in combining CAR T cells and immune checkpoint blockage (PD-1, CTLA-4, etc.) that will improve T cell persistence and anti-tumor efficacy [47]. One promising approach is to use a gene editing approach to eliminate the expression of immune checkpoint proteins (PD-1, CTLA-4, etc.), thereby making them less susceptible to the tumor microenvironment. 

Targeting immunosuppressive cells such as Tregs (with anti-CD25 antibodies) and MDSCs (anti-Gr-1 antibodies) in the tumor microenvironment may enhance anti-tumor immunity. CD33 is also present in MDSC; so, targeting CD33 will mediate anti-tumor activity through direct cytotoxicity of CD33+ blasts and also through inhibition of CD33+ MDSCs [139]. In an AML murine model, the depletion of Tregs increased the proliferation and activity of adoptively transferred tumor reactive cytotoxic T cells [140]. Lymphodepleting chemotherapy prior to CAR T cell infusion suppresses Tregs and augments the expansion of adoptively transferred CAR T cells [141]. This was previously shown in CD19 CAR T cell models through the downregulation of indoleamine 2,3-dioxygenase (IDO), a protein able to deplete tryptophan and other metabolites that inhibit CAR T cell function [142]. Therapies targeted towards the adenosinergic pathways (antibodies targeting CD73 and CD39) have proven anti-tumor efficacy in mice models [143]. A combinatorial approach targeting CD73 and IDO could potentially enhance the AML CAR T therapy; in this regard, our group is integrating universal off-the-shelf CLL-1 CAR T cells and several nanocarriers to deliver CD73 short hairpin (shRNA) and miRNA-135 as promising strategies for targeting CD73 and IDO, respectively. Combining CD73- and IDO-targeted therapy with the CAR T cell approach could potentially enhance the anti-tumor immune response by blocking two separate pathways of immune suppression.

### 3.9. Allogeneic Hematopoietic Stem Cell Transplantation with CAR T Cells

When and how to combine allo-HSCT with adoptive immunotherapy in AML is still being debated. The mechanisms of resistance to T cell-mediated anti-tumor effects after allo-HSCT are well defined in sophisticated murine models of allo-HSCT [56]. Combining a novel myeloablative irradiation-based conditioning regimen with regulatory and conventional T cell immunotherapy in haploidentical transplantation was shown to eradicate AML [144]. The published studies on how and when to combine CAR T cells in the setting of allo-HSCT have shown conflicting results. Pan et al. reported that allo-HSCT following CD19 CAR T treatment improved event-free survival and reduced relapse risk [145,146], while other studies failed to demonstrate a benefit [147]. Summers et al. reported the clearest leukemia-free survival in patients who had early loss of functionality of CAR T cells (due to CD19 CAR T loss of B cell aplasia) [148]. Data on the evidence of the benefit of allo-HSCT following CAR T cell therapy in AML patients are scarce. Zhang et al. demonstrated that among six patients who received allo-HSCT following anti-CLL1 CAR T cell treatment, four of them achieved CR [7]. One other approach took advantage of the myelosuppression effect of CD123 CAR T cells and administered donor-derived CD123 CAR T cells as a part of a conditioning regimen for haplo-HSCT [149].

## 4. Summary and Conclusions

The tremendous advances in understanding the molecular and cellular mechanisms of AML have made it possible to manipulate the immune system and BM niches. Treating AML with CAR T cells is still in an immature stage. Experience with allogeneic stem cell transplantation, which is the most effective immune cellular therapy for AML, is guiding other directed therapies. One of the major challenges in developing CAR T cell therapy for AML is the lack of a suitable antigen that is expressed uniquely on AML cells. Identifying and isolating target antigens that are homogeneously and stably expressed in all leukemic blasts and leukemic stem cells with limited on-target off-tumor toxicity, investigating complex interactions in the AML microenvironment, and seeking a suitable cell source will all improve the fine-tuning of CARs.

Sophisticated methods for ex vivo manufacturing are now changing the in vivo dynamics and the character of the final product (Figure 1). In AML, personalization should be taken a step further in directed cellular therapies with platforms that will standardize the optimal CAR design for the target antigen or antigens in line with patient-specific immunophenotyping findings, the selection of a compatible carrier cell, and the cellular subtype.

## Figures and Tables

**Figure 1 cancers-15-02713-f001:**
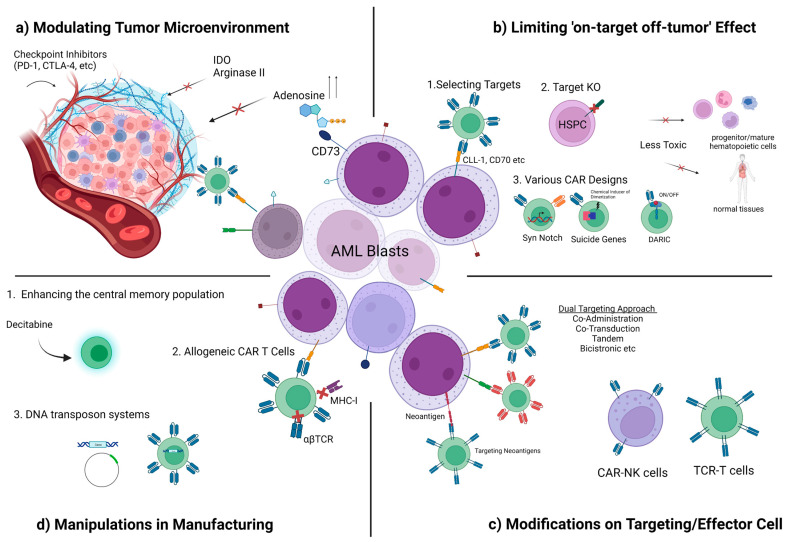
Summary of overcoming challenges for CAR T cell therapy in AML: (**a**) modulating tumor microenvironment with checkpoint inhibitors, blockage of IDO, Arginase II, and Adenosine; (**b**) limiting ‘on-target off-tumor’ effect by selecting more specific targets (such as CLL-1, CD70, etc.), knocking out targets in HSPC, application of SynNotch Systems, suicide genes, and DARIC; (**c**) combinatorial targeting, CAR NK cells/TCR-T cells; (**d**) manipulations in manufacturing such as enhancing the central memory population by decitabine, allogeneic CAR T cells, or various DNA transposon systems to carry genetic cargos. DARIC: dimerizing agent–regulated immunoreceptor complex; HSPC: hematopoietic stem and progenitor cells; IDO: indoleamine 2,3-dioxygenase (IDO); KO: knock-out; MHC-I: major histocompatibility complex-I; PD-1: programmed cell death-1; TCR: T cell receptor).

**Table 1 cancers-15-02713-t001:** CAR T cell clinical trials and case reports on AML (ALT: alanine aminotransferase; AST: aspartate aminotransferase; CLL-1: C-type lectin-like molecule-1; CR: complete response; CRS: cytokine release syndrome; Cy: cyclophosphamide; Flu: fludarabine; ICANS: immune effector cell-associated neurotoxicity syndrome; MRD: minimal residual disease; PR: partial response; SD: stable disease).

Study	Target	CAR T Cell/Lymphodepletion	Number of Patients	Response	Safety	Reference
Phase I/II	CLL-1	0.35−1 × 10^6^ kg/anti-CLL-1-CD8-41BB/Cy+Flu	8	4/8 morphological leukemia-free, MRD (−) 1 morphological leukemia-free, MRD+, 1 CR with incomplete hematologic recovery MRD(+), 1 PR, 1 SD	CRS: 5 grade 1, 3 grade 2	[7]
Phase I	CD123	Dose escalation 50 × 10^6^−200 × 10^6^/anti-CD123-IgG4-CD28/Cy+Flu	6	4/6 CR 2 reduced blasts	CRS: 4 grade 1, 1 grade 2; 1 adenoviral pneumonia requiring intubation; and 1 grade 3 rash due to drug hypersensitivity	[8]
Phase I	CD33	CD33-4-1BB/0.3 × 10^6^/kg	3	0/3 response	2 CRS; 1 ICANS. A grade 3 tumor lysis syndrome-acute kidney injury, grade 2 mucositis, grade 1 tachycardia for 1 patient; and a second patient experienced grade 2 intermittent orthostatic hypotension, grade 2 increased bilirubin and grade 3 increased ALT and AST	[9]
Case report	CD33	CD33-4-1BB/1.12 × 10^9^	1	Disease progression at week 9	Grade 4 chills and a high fever, pancytopenia	[10]
Phase I	CD38	NA	6	66.7% of patients (4/6) CR (including 1 with CR and 3 with CR with incomplete count recovery (CRi)) and full donor chimerism)	Five patients presented mild CRS (Grade I–II), and only one experienced grade III hepatotoxicity with elevated serum transaminase and bilirubin levels	[11]
Phase I	LeY	Anti-LeY-CD28/Flu-Cy	4	In the patient with active leukemia, a temporary reduction in peripheral blood blast cells was observed. One other patient achieved a cytogenetic remission, while the other two patients had SD	One patient (patient 2) had a transient grade 2 neutropenia	[12]

**Table 2 cancers-15-02713-t002:** Characteristics of various AML target molecules (ADP: adenosine diphosphate; CLL-1: C-type lectin-like molecule-1; CTL: cytotoxic T cell; DC: dendritic cell; HSC: hematopoietic stem cell; LSC: leukemic stem cell; NK: natural killer; TNF: tumor necrosis factor).

Target Antigen	Function	Expression on Normal Cells	Expression on HSCs	Expression on LSCs
CLL-1	Glycoprotein, Transmembrane receptor	Myeloid, lung, epithelial cells	-	+
CD 33	SIGLEC family protein, Transmembrane receptor	Progenitor, myeloid, Kupffer cells	+	+
CD 7	Ig superfamily/Glycoprotein, B and T cell lymphoid development, Transmembrane protein	T, NK cells, and myeloid progenitor	-	+
FLT3	Type III cytokine receptor, Tyrosine kinase receptor	Neurons, testis	+	+
CD 38	Glycoprotein, Cyclic ADP ribose hydroxylase	B, T, NK cells	-	+
CD 123	Type I cytokine receptor of IL-3, IL3 receptor subunit	Myeloid progenitors, DC, and basophils	+	+
CD 44v6	Glycoprotein, Transmembrane receptor	Keratinocytes	-	+
LeY	Glycosphingolipid, Blood group Ag	Intestinal epithelial cells	+	+
NKG2D	C-type lectin-like receptor protein, Activator receptor	NK, NKT, Tαδ, Th, and CTL	-	+
CD 70	Glycoprotein from the TNF family, Transmembrane receptor	T and B cells	-	+
CD 96	Member of immunoglobulin superfamily, adhesion of activated T and NK cells	T cells and NK cells	-	+

**Table 3 cancers-15-02713-t003:** Clinical Studies of TCR-T cells against AML (CML: chronic myeloid leukemia; CRS: cytokine release syndrome; MDS: myelodysplastic syndrome; TCR-T: T cell receptor T cell; WT-1: Wilms’ tumor-1).

Study	TCR-T Therapy	Study Phase/Number of Patients	Study Outcomes	Adverse Events
NCT02550535	Autologous WT1 TCR-T cells	Phase I/II, 10 patients (6 AML, 3 MDS and 1 TKI-resistant CML)	All 6 AML patients were alive at last follow up (median 12 months; range: 7–12.8 months). The 3 patients with MDS had a median survival of 3 months (range: 2.1–3.96 months). Two died from progressive disease and one from other causes. Two patients had disease progression.	1 CRS
UMIN00001159	Autologous WT1 siTCR-T cells	Unknown	Two patients showed transient decrease in blast counts.	None
NCT01640301	Allogeneic WT1 TCR-T cells	Phase I/II, 12 patients	With a median follow-up of 44 months (range: 21–57 months) following infusion, all 12 patients did not have evidence of disease.	None
NCT03503958	Autologous PRAME TCR-T cells	Phase I	Not posted	Not posted
NCT01621724	Autologous WT1 TCR-T cells	Phase I/II, 7 patients	Not posted	Not posted

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
