# Peer review of "The Black Hole: CAR T Cell Therapy in AML"

_cancers, 2023, doi:10.3390/cancers15102713_

Round 1

Reviewer 1 Report

In this manuscript, Erdan and Karim reviewed the current progress on CAR T cell therapies, which are very promising immunotherapeutic options for the treatment of patients with AML. This review focuses on not only the challenges of applying/developing CAR T cell therapies but also the potential strategies to overcome these difficulties. This review also discussed other immunotherapeutic options, such as tailoring the tumor microenvironment, targeting neoantigens and allogeneic approaches, which are of clinical importance.

Overall, this review is well-written and provides important insights for future studies that focus on advancing CAR T cell-based therapies in AML. To strengthen the manuscript, the authors should make efforts to address the following concerns.

Comments:

1.     References are missing in the introduction section (lines 37-45).

2.     Table 1 is cut in half. Authors should place the entire table on one page.

3.     “CD123, which has the IL-3 receptor subunit” in line 143 should be “CD123, which is the IL-3 receptor subunit”.

4.     Is better to move “Cooper and colleagues developed a lentiviral DARIC construct that targets a C2 splice isoform with the membrane proximal domain of CD33, and a Phase I study using this strategy is now open for enrollment [72].” In lines 264-267 to line 261 in front of “We have recently…”

5.     In line 288, the statement of “(CD33+CD123-, CD33+CD123-)” is incorrect.

6.     “A universal CAR T cell platform (on/off switching mechanism) successfully targeted CD33 and CD123 AML blasts in vitro and in vivo [84].” in lines 297-298 is not a sentence.

7.     It is better to have a short description for the figure (page 12) rather than putting just the abbreviation as the figure legend.

8.     All tables and figures should have their own titles.

9.     "Author contribution" in line 491 should not belong to the figure legend.

Reviewer 2 Report

Authors have written a quite comprehensive review regarding CAR T cell therapy in AML. The major points have been well explained and suggestion of including more sophisticated next-generation CAR T, TCR-T and CAR-NK therapies; approaches to tailor the microenvironment and target neoantigens; and allogeneic approaches is quite acceptable. 

Moderate editing of English language is required.

Reviewer 3 Report

The authors present a comprehensive review of the field of cellular therapy for AML. The manuscript is well-organized and well written. 

My main issue the paper is the title. The authors use the term "black hole" as a catchy phrase, however I am having difficulty understanding how immune therapy for AML has anything in common with a black hole. A better analogy for immune therapy in AML might be dark matter- something we know is there but are having difficulty defining and understanding.

Line 97: the authors should also mention DDX41. 

There are grammatical errors throughout. There is an occasional typo (spacing in the table). 
